# Archaeology and contemporary death: Using the past to provoke, challenge and engage

Karina Croucher[1]*, Lindsey Büster[1,2], Jennifer Dayes[3], Laura Green[4], Justine Raynsford[5], Louise Comerford Boyes[6], Christina Faull[7]

1 School of Archaeological and Forensic Sciences, Faculty of Life Sciences, University of Bradford, Bradford, United Kingdom, 2 Department of Archaeology, University of York, York, United Kingdom, 3 Department of Psychology, Faculty of Health, Psychology and Social Care, Manchester Metropolitan University, Manchester, United Kingdom, 4 Nursing, Midwifery and Social Work, Faculty of Biology, Medicine and Health, University of Manchester, Manchester, United Kingdom, 5 Faculty of Health Studies, University of Bradford, Bradford, United Kingdom, 6 Division of Psychology, School of Social Sciences, Faculty of Management, Law & Social Sciences, University of Bradford, Bradford, United Kingdom, 7 LOROS Hospice, Leicester, United Kingdom

* k.croucher@bradford.ac.uk

**Data Availability Statement:** All relevant data are within the manuscript and its Supporting Information files.

## Abstract

While death is universal, reactions to death and ways of dealing with the dead body are hugely diverse, and archaeological research reveals numerous ways of dealing with the dead through time and across the world. In this paper, findings are presented which not only demonstrate the power of archaeology to promote and aid discussion around this difficult and challenging topic, but also how our approach resulted in personal growth and professional development impacts for participants. In this interdisciplinary pilot study, archaeological case studies were used in 31 structured workshops with 187 participants from health and social care backgrounds in the UK, to explore their reactions to a diverse range of materials which documented wide and varied approaches to death and the dead. Our study supports the hypothesis that the past is a powerful instigator of conversation around challenging aspects of death, and after death care and practices: 93% of participants agreed with this. That exposure to archaeological case studies and artefacts stimulates multifaceted discourse, some of it difficult, is a theme that also emerges in our data from pre, post and follow-up questionnaires, and semi-structured interviews. The material prompted participants to reflect on their biases, expectations and norms around both treatment of the dead, and of bereavement, impacting on their values, attitudes and beliefs. Moreover, 87% of participants believed the workshop would have a personal effect through thinking differently about death and bereavement, and 57% thought it would impact on how they approached death and bereavement in their professional practice. This has huge implications today, where talk of death remains troublesome, and for some, has a near-taboo status–'taboo' being a theme evident in some participants' own words. The findings have an important role to play in facilitating and normalising discussions around dying and bereavement and in equipping professionals in their work with people with advanced illness.

**Funding:** The project 'Continuing Bonds: Exploring the meaning and legacy of death through past and contemporary practice' was funded by the Arts and Humanities Research Council, Grant Number AH/M008266/1. The Principal Investigator was Karina Croucher, and Co-Investigators were Christina Faull and Laura Green. https://ahrc.ukri.org/ The funders had no role in study design, data collection and analysis, decision to publish, or preparation of the manuscript.

**Competing interests:** The authors have declared that no competing interests exist.

# Introduction

As a result of advances in healthcare and changing cultural preferences, most people in the developed world die in hospital [1], although the majority of individuals wish, in fact, to die at home [2–5]. Furthermore, the dying and their families are often unprepared for decisions regarding preferences for care when health declines and death is potentially imminent [6, 7]. A lack of willingness to talk about the realities of death and dying amongst healthcare professionals has complex derivation, but is underpinned by a lack of reflection upon the cultural contexts which influence the delivery of palliative and post-mortem care, as well as a lack of confidence and skills in approaching patients and families with a topic that is perceived as having the potential to cause harm [8, 9]. For some, this means that discussions around death are avoided [10, 11] although Walter [12] notes that this may be contextual, dependent on social situation and individual preference. However, fear of death remains powerful, and is often mediated by cultural and religious beliefs and rituals [13]. There remains a lack of research and literature into attitudes towards death and grief explicitly among healthcare professionals in the UK [14] although surveys on perceptions in the general public highlight the need for more opportunities for dialogue about death. For instance, an Ipsos MORI poll in 2019 of a representative sample of 966 British adults participating in a series of public workshops around art installation 'the departure lounge', agreed with the statement that 'death is still not discussed and palliative care not well understood' [10, p4] although all participants welcomed the opportunity to discuss death and called for more spaces for discussion. Death, while perhaps no longer 'taboo', remains 'troublesome' for many to discuss, at least in terms of opening dialogues and starting conversations, a topic brought to light by the Dying Matters Coalition in the UK, and discussed by Walter [15, 16] who recognises that conversations around death are difficult and frequently 'shut down', rather than taboo. Most recently, the Covid-19 pandemic has presented a highly visible spectre of death and it remains to be seen how this may change societal willingness to have discussions and debate; media reports so far suggest a rise in the demand to discuss death in the pandemic environment [17, 18]. Paradoxically, as a consequence of medical advances, living with chronic illness for some time before terminal decline has become the norm, and death has become arguably easier to anticipate: around 80–90% of deaths are expected [19, 20]. Despite this anticipation, death is easier to talk about in the abstract, rather than when related to the foreseeable death of a loved one. For some, while death is fascinating, this often does not translate into direct personal action, for instance, in writing wills [21] or planning for care and choices in advance, suggesting there remains a barrier between broader discussions around death vs individual direct personal contemplation [15]. A survey of over 2000 people in Britain with cancer revealed that 35% had not shared with anyone their thoughts and feelings about dying, with only 8% speaking to their healthcare team [22]. A study based on semi-structured interviews suggests that when physicians feel more comfortable about death, they are more likely to feel comfortable talking to their patients about it [23]. The Cooperative Funeral care providers recommend innovative approaches to encouraging conversations around death, dying and bereavement [21]. Palliative care research has also called for ways to address the problematic status of the topic of death and dying in society in order to create an environment for more 'good deaths', defined as 'deaths with dignity and aligned with patient's preferences for care and after death care' [14, 24–27]. Discussions regarding death are crucial to enable more positive experiences, as is the acceptance of dying as a normal part of the life cycle, rather than viewing death as medical failure. There have been many calls for normalising discussions surrounding death: by the End of Life Care Strategy [28] and the More Care Less Pathway review [29] of the Liverpool Care of the Dying Pathway in the UK, and internationally by the World Health Organisation [30, 31]. Instigating

conversations early can help with planning, enabling easier dialogue before a terminal diagnosis or bereavement [32] however, for many, thoughts only turn to death when they are facing bereavement at a time when conversations are even more challenging [21].

The 24-month project, 'Continuing Bonds: exploring the meaning and legacy of death through past and contemporary practice' (funded by the UK-based Arts and Humanities Research Council; 2016–2018), explored a new and creative approach to addressing this need. Using archaeology, it provided an alternative framework for reflecting on death, through the provision of accessible archaeological case studies that illuminated the variety of methods through which past societies have dealt with the dead. Moreover, it focused on one particularly enduring concept: that of 'Continuing Bonds' between the living and the dead. The idea of Continuing Bonds has been developed by sociologists and psychologists as a framework for understanding the grieving process, where it is understood that death is not the end of the relationship between the living and the dead, but that the dead continue to have meaning and significance in the lives of the living [33–35].

An event in 2014, '*what will survive of us. . .*' examined how the discovery of the Plantagenet King, Richard III, beneath a carpark in Leicester could act as a catalyst for the exploration of death practices and 'continuing bonds' with ancestors, including the spiritual dimensions and complexities of re-interment. This led to an exhibition (with Professor Sarah Tarlow, University of Leicester) displayed in public spaces in Leicester in 2015. During the exhibition, project members witnessed a transition in conversations as members of the public talked about their own experiences and expectations as a direct result of engaging with the archaeological material; this observation further inspired the development of the Continuing Bonds Project and our aim to explore how archaeology can both reveal and inform attitudes to death and dying.

The Continuing Bonds project comprised an interdisciplinary team of archaeologists, end of life care professionals and a counselling psychologist/qualitative psychology researcher, with support from experts in pharmacy, sociology and social psychology. The team were based across universities and hospices in two UK cities: one in the North of England and one in the Midlands.

This was the first project of its kind (to the knowledge of the authors) that explicitly used archaeology with end-of-life care practitioners and students. Subsequent projects have aligned with our aims (e.g. [36, 37]). The role played by museums in enabling discussions around death through displays of the dead has been discussed by Giles and Williams [38]. Tarlow and Sayer have been prolific in discussing the value of archaeology in discussions of mortality [39, 40]. Furthermore, there is a growing recognition of the value of arts and humanities to societal wellbeing [41, 42] and more broadly, positive impacts on mental health (e.g. [43–46]). Such projects are establishing the value that archaeology and heritage add to social and mental wellbeing.

Our project aims were threefold: firstly, to explore the value of archaeology for facilitating discussion around death and bereavement; secondly, to enable reflection on contemporary attitudes to death and bereavement, challenging norms and biases; and thirdly, to understand the potential impact of workshop attendance on individuals, professional practice and patient care.

This paper outlines our findings, along with a reflection on why and how we believe the project succeeded.

## Materials and methods

The Continuing Bonds project involved delivering workshops to health and social care professionals and students. The workshops showcased archaeological materials, usually through

printed pictures and accompanying text, along with audio-visual resources and artefacts. The archaeological case studies sought to challenge perceptions of what many of our participants might consider to be 'normal' or 'usual' treatment of the dead.

The project used mixed methods to explore the impact of the workshops. An action research approach [47] was adopted, in which the method (both of workshop facilitation and data collection) iteratively evolved in recognition of the findings and themes emerging through ongoing analysis.

## Ethical approval

Ethical approval was awarded by the University of Bradford (E487) and from the NHS (IRAS 203842). The project was registered on the NIHR portfolio for research (CPMS 33182). Informed consent was gained from all workshop participants to use their data for research and publication. All individuals received information sheets prior to participation, which included potential risks, they were alerted to the use of images of human remains (such as might be viewed in museums), and they were provided with sources of further information and support. Workshops were facilitated to ensure ethical compliance, including 'ground rules' for each workshop (such as confidentiality and respect), and each workshop included facilitators trained in psychological/emotional support.

## Participants

Health and social care professionals and students in Leicester and Bradford were recruited through local organisation emails, flyers and posters. Advertising utilised hospices, local bereavement support (e.g. CRUSE; Bradford Bereavement Support), universities, hospitals and GP surgeries, as well as websites and social media. Individuals were invited to participate in as many workshops as they wished.

## Archaeological materials

Eight workshop themes were chosen, with bespoke contents for each theme. The first four themes were outlined in the project proposal: 'memorialisation and legacy' (Fig 1A); 'age and circumstance of death (Fig 1B)'; 'images of the dead'; and 'ancestors'. Consistent with our action research approach [39], the subsequent four themes were then developed from our analysis of the first round of participation; these were 'place' (Fig 1C); 'objects and the dead' (Fig 1D); 'legacy'; and 'treatment of the dead'.

The themes included several sub-themes, each comprising between one and four archaeological/ethnographic case studies (archaeological examples are from the past; ethnographic examples are from contemporary societies in the UK and around the world), designed to illustrate a breadth of perspectives (S1 Table). Themes were grounded in current research and debates in the archaeological literature [48]. They were also chosen to highlight different attitudes to death, dying and bereavement, encouraging participants to compare and contrast the materials from different perspectives (e.g. individual vs communal, whole vs fragmented, past vs present). Materials frequently contrasted archaeological examples with images and case studies from a wide variety of global contexts, including the contemporary world. For example, one set of materials contrasted a Victorian Death Portrait (which included a photograph of a deceased child) with a family portrait taken on a smartphone of relatives posing with the mummified body of their grandmother, as part of funerary celebrations in modern-day Sulewesi, Indonesia (see [48] for further discussion on a selection of the archaeological case studies used). The case studies needed to be visually striking, convey information succinctly and have the potential to challenge thinking and help participants question their pre-existing attitudes

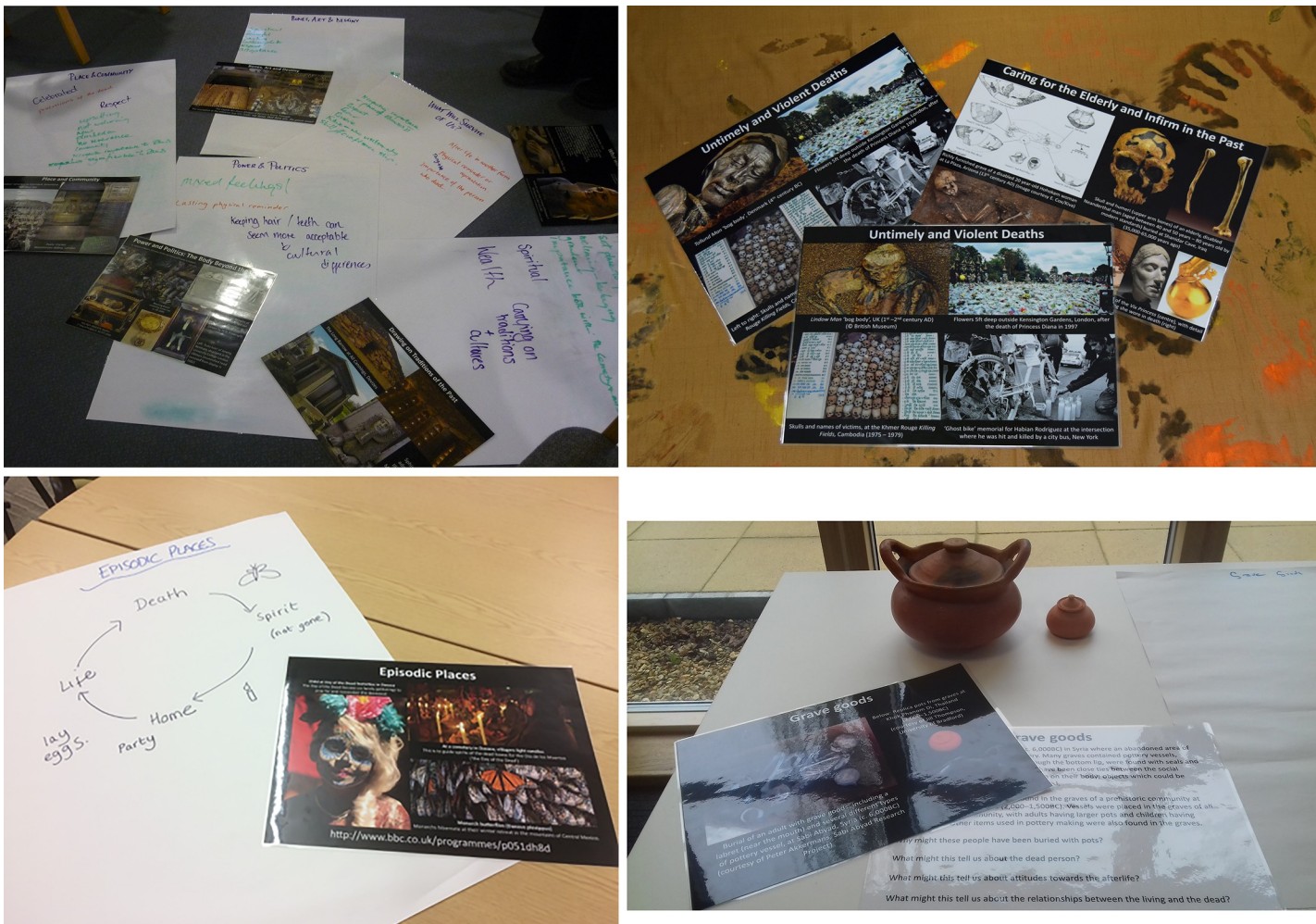

**Fig 1. Examples of workshop station materials.** A) Images and participant group notes on materials in a 'memorialisation and legacy' theme workshop. B) Images from an 'age and circumstance of death' workshop. C) Images and participant notes from a 'places' workshop, D) Replicas of pottery vessels recovered from graves in an 'objects and the dead' workshop.

and experiences. It was intended that participants' initial reactions to an image would be challenged by details presented in the information on its reverse, thus revealing unconscious cultural biases. For instance, in the 'Place' theme, text added contextualisation to the image of disarticulated human remains found in the Sculptor's Cave funerary site (Scotland, UK< 11th-9th centuries BCE), moving the viewer from the assumption of violence, to possibilities of compassion, in what appear at first sight to be brutal practices. Case studies were arranged, where possible, in order of perceived 'accessibility', with participants starting with the most 'familiar' topics and moving onto progressively more 'challenging' materials. For instance, in the workshop on the theme of 'Memorialisation and Legacy' (Fig 1A), participants would view materials on a statue of Rameses II from Luxor, Egypt and a poem by the 19th century poet Percy Shelley, before seeing images of Long Barrow Neolithic tombs and a monumental Victorian/Edwardian cemetery in Bradford. The final station introduced the topic of 'the body beyond the grave' with images of saints' replicas, the preserved body of Jeremy Bentham in London, and the architectural chapel of bones, 'Capella dos Ossos', in Portugal. For the 'Objects and the Dead' workshop, the topic of 'Heirlooms' was most accessible, showing a 6th century BCE belt

plate which had been broken, repaired and reused through time, and a necklace from an Arras burial in Yorkshire, UK (c.400-200 BCE), which had been made of around 100 beads from potentially 5 different necklaces. The next station discussed objects placed directly into graves, 'Grave Goods', including images of a burial with pottery from a prehistoric site in Syria and replicas of pottery recovered from Khok Phanom Di, Thailand (2000–1500 BCE) (Fig 1D). The final station showed images of 'Human remains as objects', including a spindle whorl made from a human femur (Scotland, UK, AD 130–340), and a painted human mandible worn as a pendant (Mexico, c.AD 700). A variety of media were presented in the workshops; primarily the resources comprised printed photographs with accompanying text, and some case studies presented as video clips. Physical objects were also occasionally included, such as a facial recreation of the Bronze Age skull of 'Gristhorpe Man' and replicas of pottery vessels recovered from graves (Fig 1D).

## Workshop design and delivery

A total of 31 workshops were delivered, each of the eight themes ('memorialisation and legacy'; 'age and circumstance of death'; 'images of the dead'; 'ancestors'; 'place'; 'objects and the dead'; 'legacy'; and 'treatment of the dead') being repeated least three times. Each two-hour workshop was facilitated by three team members using a semi-structured approach. After introductions to the project, confirming consent, and instructions, participants were assigned to a group of two-five people, which moved around a series of 'stations', each addressing a different aspect of the workshop theme. For instance, in the workshop on 'Objects and the Dead', the stations addressed the sub-themes of 'Heirlooms', 'Grave Goods' (objects placed in the grave), and 'Human remains as objects'. Each group spent time at every station. As the project adopted an action research ethos, the design and delivery of subsequent workshops iteratively evolved in response to participant feedback and team observations; stations reduced in number and content, enabling deeper engagement with the case studies.

Following small group discussions at each station, participants were then brought together for a focus group discussion, facilitated by two of the project team. The focus groups had a semi-structured format, which began by asking participants which materials prompted the most discussions; in many cases, little facilitation was needed as discussion arose organically from participants. Focus group discussion was brought to a close by asking the participants for any reflections on the potential impact of the workshop experience on their understanding and professional practice.

At the end of each workshop, the wellbeing of the group was checked. Support networks and follow-on materials were signposted. Participants were then invited to take part in a quick art activity, using paint to add their handprint to a communal art piece. This piece not only echoed some of the earliest art left by prehistoric people, but the handprints themselves, being both highly individual yet anonymous, symbolised the personal nature of the accounts contributed by workshop participants in this study (Fig 2). This light-hearted activity of individual mark making also served as a transition point between discussions in the workshop and the restoration of personal/professional activities in the outside world.

## Data collection

To provide evidence as to the impacts attributable to participation, quantitative and qualitative data were collected. Firstly, participants were asked to complete questionnaires at the beginning (Pre-WQ), end (Post-WQ) and 1–3 months after attendance at a workshop in a Follow-up Questionnaire (FUQ). Questionnaires included a mix of Likert scale questions and free text

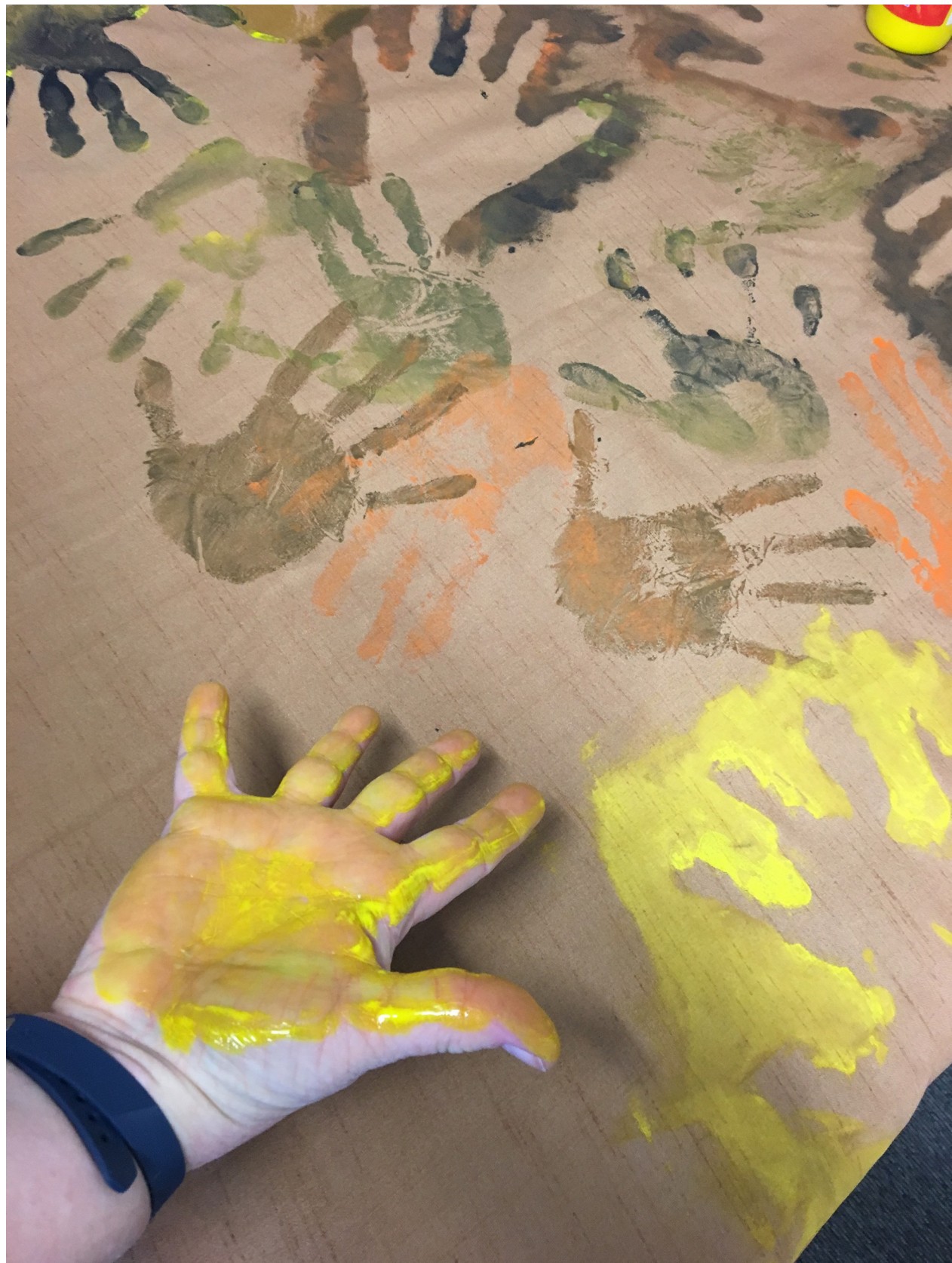

**Fig 2. Closing activity: Handprint artwork.**

responses. Individuals who attended multiple workshops were asked to complete a Post-WQ on each occasion.

The Pre-WQ (S1 File) collected demographic information along with the degree and nature of participants' experience of working with people at the end of their lives.

The Post-WQ (S2 File) sought to establish the participants' extent of agreement to the following statements:

- The workshop made me think about death, dying and bereavement in different ways;

- Prior to the workshop, I felt confident in talking about death, dying and bereavement with family, friends, peers and patients;

- I anticipate that my confidence in talking about death, dying and bereavement has increased as a result of the workshop;

- The workshop will impact how I approach death, dying and bereavement in my professional practice;

- I felt more comfortable talking about personal experiences regarding death, dying, bereavement and loss in the workshop than I would in general life.

    Participants were also asked:

- To identify words and/or suggest words which most accurately described their experience of the workshop;

- And, whether they would recommend the workshop to a colleague.

The FUQ (S3 File), which was emailed to participants, included a mix of categorical and free text responses seeking reflection on the workshop and any impact experienced in personal life and professional practice.

In addition, eleven survey respondents were then interviewed, around 6–9 months after participation, with semi-structured interviews being employed to qualitatively explore participant sense-making of and meanings attributed to participation. Interviewees were purposively selected to gain a diversity distribution based on professional/educational background (including whether a student or qualified), gender, workshop topic attended, number of workshops attended and workplace setting. They were sampled from the 49 individuals who volunteered to have an interview in their FUQ response. Consent was gained prior to the interviews, and interviews were conducted via the telephone or face to face at a mutually convenient time, with the semi-structured interviews conducted by a member of the project team, and later transcribed by a reputable transcriber with a clearly agreed confidentiality policy.

## Data analysis

Quantitative data was collated using Excel spreadsheets and analysed using descriptive statistics. For each question, free text responses were analysed thematically employing an inductive, iterative approach [49]. Text was manually coded, and theme development was bottom up, driven by the content of the comments and codes. Interview transcripts were subjected to a thematic analysis [50] using an inductive approach that permitted themes to emerge through iterative cycles of analysis. All data were analysed by at least two members of the project team.

## Findings

The project recruited 146 individual participants. Twenty-four participants attended two or more workshops, with 187 workshop participations in total. All 146 participants were sent a

link to the FUQ via email, which 85 participants (58%) completed. Eleven volunteers were selected for semi-structured interviews, from those who had given permission in their FUQ responses.

There were 127 female and 19 male participants, of whom 89% were white British, and 61% described themselves as religious or spiritual. Most participants were qualified professionals (77%) including nurses (39%), counsellors/psychotherapists (25%), occupational therapists (4%), social workers (6%), physiotherapists (4%), doctors (8%) and chaplains (4%). Other professionals (11%) included researchers and speech and language therapists. Twenty-three percent of participants were students: primarily they were training in nursing (54%), followed by those studying counselling/psychotherapy (21%), health/wellbeing/social care/social work (17%), and medicine (8%). In terms of working in end-of-life contexts, 56% had substantial experience, 34% a small amount, and 11% had no experience.

Post-WQ data revealed that:

- Almost all of the survey respondents (93%) thought that archaeological materials could be used to facilitate discussions about death, dying, bereavement and loss (Table 1A);

- The majority of participants (84%) agreed that the workshop had made them think differently about death, dying and bereavement (Table 1B);

- Just under half of the participants (47%) anticipated an increase to confidence in talking about death, dying and bereavement, as a result of the workshop (Table 1D);

- Over half (57%) of participants reported that the workshop impacted upon how they would approach death, dying and bereavement in their professional practice (Table 1E);

- 41% agreed (compared with 34% who disagreed) that they felt more comfortable talking about death, dying, bereavement and loss in the workshop than they would in general life (Table 1F);

- Overwhelmingly, participants said that they would recommend the workshop to others (93%) (Table 1G).

The analysis of interview data gave rise to superordinate (or 'master' themes) that included: 'Transition'; 'Threshold'; 'Diversity and Difference'; 'Territory'; 'Taboo'; 'Signification'; 'Paradox'; 'Epiphany'; 'Discourse and Space'; 'Feelings' and 'Catalyst'. These themes are referred to and illustrated in the discussion of key findings.

Overall, the data evidences four main key findings: Archaeology can facilitate conversations around death, dying and bereavement; Archaeology challenges cultural norms and highlights diversity in death practices; the workshops influenced people in their professional practice; and the workshops were a positive experience. These findings are discussed in greater detail below, together with supporting quotations, which are referenced by workshop theme, questionnaire type, and anonymous participant number. Quotations from interviews are referenced by a randomly assigned number, along with transcript page and line number.

## Key finding 1: Archaeology can facilitate conversations around death, dying and bereavement

Of the 149 participants who were asked, 93% agreed that archaeological materials could be used to facilitate discussions about death, dying, bereavement and loss (Table 1A). Participants acknowledged that such opportunities were lacking, even within the palliative care profession: "...*many people within the health and care sectors are not given opportunities to discuss death*

**Table 1. Responses to Post-WQ.**

a) "In your opinion, do you think archaeological materials can be used to facilitate discussions about death, dying, bereavement and loss, or training in this area"

|  | N = 150* | % |
|---|---|---|
| Yes | 139 | 92.66 |
| No | 9 | 6 |
| No data | 2 | 1.33 |

b) "The workshop made me think about death, dying and bereavement in different ways"

|  | N = 187 | % |
|---|---|---|
| Strongly agree | 57 | 30.48 |
| Agree | 100 | 53.47 |
| Neither agree nor disagree | 15 | 8.02 |
| Disagree | 7 | 3.74 |
| Strongly disagree | 7 | 3.74 |
| No data | 1 | 0.53 |

c) "Prior to the workshop, I felt confident in talking about death, dying and bereavement with family, friends, peers and patients"

|  | N = 187 | % |
|---|---|---|
| Strongly agree | 76 | 40.64 |
| Agree | 82 | 43.85 |
| Neither agree nor disagree | 13 | 6.95 |
| Disagree | 12 | 6.41 |
| Strongly disagree | 4 | 2.13 |

d) "I anticipate that my confidence in talking about death, dying and bereavement has increased as a result of the workshop"

|  | N = 187 | % |
|---|---|---|
| Strongly agree | 24 | 12.83 |
| Agree | 63 | 33.68 |
| Neither agree nor disagree | 78 | 41.71 |
| Disagree | 19 | 10.16 |
| Strongly disagree | 3 | 1.60 |

e) "The workshop will impact how I approach death, dying and bereavement in my professional practice"

|  | N = 187 | % |
|---|---|---|
| Strongly agree | 18 | 9.62 |
| Agree | 89 | 47.59 |
| Neither agree nor disagree | 62 | 33.15 |
| Disagree | 15 | 8.02 |
| Strongly disagree | 2 | 1.06 |
| No data | 1 | 0.53 |

f) "I felt more comfortable talking about personal experiences regarding death, dying, bereavement and loss in the workshop than I would in general life"

|  | N = 150* | % |
|---|---|---|
| Strongly agree | 24 | 16 |
| Agree | 38 | 25.33 |
| Neither agree nor disagree | 37 | 24.66 |
| disagree | 45 | 30 |
| Strongly disagree | 6 | 4 |

g) "Would you recommend the workshop to others"

|  | N = 187 | % |
|---|---|---|
| Yes | 174 | 93.04 |

(*Continued*)

**Table 1.** (Continued)

| | | |
|---|---|---|
| No | 1 | 0.53 |
| Don't know | 9 | 4.81 |
| No data | 3 | 1.60 |

*question introduced part way through the project.

*and dying and are less able to cope with them when they encounter them. The workshop seemed to be a "safe" and accessible way of broaching the subject, with less likelihood of raising internal barriers when discussing the topic"* (Objects, FUQ, 043).

Participants felt that archaeology provided a non-threatening way to introduce the topic of death and dying, helping to remove barriers to a difficult topic, with one person stating that '*these materials are non-personal, non-judgemental and so "safe" in many ways*" (Legacy, Post-WQ, 002). Participant reflections alluded not only to the importance of 'safe' materials, but also of a 'safe space'; "*[The workshop was] well-structured and therefore [gave] safe group experiences*" (Treatment of the Dead, Post-WQ, 008), and "*I felt more comfortable because I would not expect the conversation to upset anyone as we have come together with this discussion in mind*" (Ancestors, Post-WQ, 010). Another participant commented that "*The workshop seemed to be a "safe" and accessible way of broaching the subject, with less likelihood of raising internal barriers when discussing the topic*" (Objects, FUQ, 043). Further participants indicated that this feeling of 'safety' encouraged more in-depth personal reflection, as exemplified by the comment that the materials *"can move people from academic knowledge to personal experience/ feelings etc."* (Treatment of the dead, Post-WQ, 003).

Of the participants asked, 41% strongly agreed or agreed that they felt more comfortable talking about personal experiences regarding death, dying, bereavement and loss in the workshop than they would in general life (Table 1F). Although 34% disagreed or strongly disagreed with this, 58% of those who disagreed/strongly disagreed (12% of the total number of participants asked this question) elaborated in free text comments that they already felt comfortable talking about these topics prior to the workshops.

**Conversations beyond the workshops.** The workshops prompted continued discussion about death and dying beyond immediate participation. Of the 85 respondents who completed a FUQ, 60% noted that they had discussed the workshops with colleagues. One respondent recalls a conversation during the workshop being drawn on later, writing that the workshop materials on plastered skulls had caused reflection on the use of "*dead peoples body parts or ashes to create artefacts that 'live on' and can be used"*. They stated that they remembered the conversation because of the *"staff member's honesty at sharing that she had turned her dad's ashes into a necklace which she wears."* The participant wrote about how, *"after the workshop [the staff member] purposefully brought it in to show [her] which [she found] very touching"* (Ancestors, FUQ, 53), indicating that the workshop had a lasting emotional impact on this participant also. Another participant recalled that she had "*spoken about the different ways people remember their ancestors to colleagues*" (Objects, FUQ, 173), and another shared that "*[the workshop] ha[d] been a great discussion point with [their] colleagues*" (Objects, FUQ, 153). One participant reflected that '*Workshops like this are very helpful in this context to increase the ease with which you approach the dead or the dying. They help to reduce stigma, break down boundaries and accept death as a normal part of life*' (Images of the dead, FUQ, 162).

Of the 85 respondents, 40% had discussed the workshop with family and 34% with friends. One participant, for example, described a conversation around their own post-mortem wishes: "*I visited my mum's grave on Sunday for Mothers' Day and my son and I talked about how we want to be*

*buried. He wants to be under a tree. I told him about the jaw bone on a necklace that I saw with you"* (Objects, FUQ, 034), while another participant wrote that '*I've told friends about what I learned and considered attitudes to death in my own culture and family. I was most struck by the care taken in ancient societies over burying the dead and by the difference in attitude to images of the dead between some cultures I saw and my own*' (Age and Circumstances of Death, FUQ, 199). Only 18% of FUQ respondents had not discussed the workshop with family, friends or colleagues.

The workshops had the additional impact of empowering participants to take steps in their own personal preparations for dying, illustrated by reflections such as "*This type of workshop would not only go a long way in raising awareness about death, dying and bereavement but [would] also make. . . people think about how they would want things done when they die*" (Objects, FUQ, 035). Other participants stated that they had discussed the workshops '*with work colleagues* ' (Objects, FUQ, 132), and '*It has made me think more about my wishes when I die and also think about what my children should do after I have gone with regard to mourning period and rituals*' (Objects, FUQ, 173).

The interview analysis theme of 'Discourse and Space' evidences how much the chance to talk and a creative, supportive space in which to do so was valued. This is not limited to reflections on the efficacy of the workshops, e.g. subtheme 'Unusual Rites Prompted Better Discussion' but also that discussion of personal experiences in a safe space was valuable to the individual. The workshops also supported a professional skills gain: e.g. subtheme 'Difficult Conversations More Easily Had'.

The theme of 'Signifiers': ritual, the presentation and representation of the dead, the use of physical artefacts in memorialisation practices but also psychological signifiers (e.g. the release of butterflies to help the deceased to transition, the placing of bones in caves) is strongly linked to the theme of 'Territory' with its connotations of ownership and boundaries: whose death is it, after all? These can be difficult conversations that are as much about the needs of the living as they about the dead. There was an interesting amount of interview data about societal tendencies to plan a good remembrance of the dead, rather than plan a good death for the dying. Here a tension can be evident in terms of who owns the death: e.g. subtheme 'My Death Is, and Isn't, About Me', the rights of the dying vs the rites of the dead.

'Taboo' also features thematically in the data, not only in terms of death being something whereby more taking and openness would perhaps help ameliorate the difficulties surrounding the phenomenon, but there being some nuance to this–some participants felt that expressing relief about someone dying is still taboo, perhaps in part due to how hard it is to articulate the concurrent emotions of profound grief of loss and profound relief of release. One participant discusses the safe space of the workshops in the context of being challenged, talks about '*The culture of the workshop and the skills of the facilitators and the safety there was in the room that people are able to talk about these issues, these difficult issues, and look at images that might have been perhaps disturbing in a safe way. And that's a relief actually because, as you know better than I, it's one of the last taboos*' (P5, p2, 68–71).

## Key finding 2: Archaeology challenges cultural norms and highlights diversity in death practices

Table 1B shows how participants responded to the workshops, indicating that 83% considered participation to have made them think differently about death, dying and bereavement. Their free text responses illustrated how this happened:

Participants told us that *"Cross cultural approaches allow us to question our assumptions when faced with very different perspectives from other times and places"* (Legacy, Post-WQ, 012), showing how the materials encourage self-reflection around personal beliefs.

*"The images from the workshop were very powerful and thought provoking and reinforced the different attitudes towards death in different cultures and at different periods in time"* (Ancestors, FUQ, 013), revealing the diversity of practices and the culturally-contextual nature of reactions to death, and encouraging acceptance of difference: *"hearing about death and burial rituals in different cultures throughout history. . . "gives permission" for people today to react to death in their own way"* (Objects, FUQ, 016). Another participant said that the archaeological materials reinforced that *'. . .we are all trying to find meaning no matter what belief system or race"* (Ancestors, FUQ, 017).

The comments indicated that it was often the more challenging case studies (those where material was particularly removed from familiar experiences) which led to the biggest reactions: *"[I particularly remember] the case that highlighted the cultural tradition of exhuming dead bodies as a means to celebrate and remember their life. I remember this because it is so far removed from how we culturally behave and our group generally found it quite a shock which generated a good deal of discussion. . ."* (Ancestors, FUQ, 018).

Some participants noted that it was not only the archaeological materials which stimulated a challenge to beliefs and reflection but also the workshop participants and the group working: *"I found the day interesting to see the differing attitudes and level of knowledge other allied healthcare professionals had about death and dying"* (Objects, FUQ, 020).

Other comments from participants suggested the use of the workshop for broader training around diversity, with one participant stating that *"I think they [workshop materials] would help widen understanding of different approaches to death, dying and bereavement. Perhaps more effective than traditional equality and diversity training"* FUQ44.

It is no surprise that 'Diversity and Difference' was a strong theme in the analysis of interview data. Being designed to support reflection, the workshops offered physical, mental and emotional space allowing participants to explore and question, to better appreciate 'strange-to-me' and unfamiliar practices, to glimpse the cultural appositeness of the unfamiliar, ultimately to rethink and reframe practices closer to home for their 'rightness' and inevitability. Thus, the impact of the workshops goes beyond participants recognising obvious physical and material variation in physical rites and extends to reflecting on how norms, values and traditions are culturally invested and embedded. If phenomena are thus socially constructed—conditional and to which we are conditioned–then there is room for change, personal development, transition.

## Key finding 3: The workshops influenced people in their professional practice

Overall, 58% (n = 96) of participants agreed or strongly agreed in the post-workshop questionnaire that attendance at the workshop would influence the ways in which they approached death, dying and bereavement in their professional lives (Table 1E).

Furthermore, a high proportion of the student participants, 21 out of 24 (87.5%), agreed that the workshop would impact upon their practice.

The free text comments from the Post-WQ and the FUQ provide further details on this impact and examples of how it played out in their work: *"I am beginning to open up when discussing dying at the hospice. . ."* (Memorialisation and Legacy, FUQ, 027), and *"I have mentioned the workshop as a way of bringing dying up in conversation with families, when the time is right of course"* (Objects, Post-WQ, 026). Another participant commented that '*I referred to it with a family member/service user talking about how we are important to those left behind when we have died"* (Age and Circumstances of Death, FUQ, 029).

For some, the impact related to general confidence, knowledge and awareness of diversity in death practices, with one participant reflecting that *"I will now feel more aware of the history*

*of my patients and how to approach the subject with them"* (Age and Circumstances of Death, Post-WQ, 021), and another stating that *"I give talks on the work we do and it has probably made it easier for me to be bolder and braver in talking about bereavement"* (Objects and Images of the Dead, FUQ, 024).

Prior to the workshop, 84% (n = 158) of participants felt that they were confident in talking about death, dying and bereavement with family, friends, peers and patients; this included 79% of students, with only 8.5% (n = 16) disagreeing/strongly disagreeing. Despite the majority of participants reporting high confidence levels prior to the workshop, 47% (n = 88) (including 42% of students) anticipated an improvement in their confidence to have such discussions after attending the workshop. Of the 16 participants who were not confident prior to the workshop, 68% agreed/strongly agreed that their confidence would be improved as a result of workshop participation.

Whilst the project sought to explore professional changes and asked explicit questions about this in both the Post-WQ and FUQ, the data showed that many participants also reflected on their own personal beliefs and experiences, and in many cases, it was difficult to distinguish between personal and professional impacts: *"This has interested me through thirty years of practice and has given me a chance to refresh, consolidate and rethink both my work and my personal responses"* (Objects, FUQ, 031); *"The workshop has helped me to "normalise" thinking on the issues around death and dying, both professionally and personally. It has been most helpful"* (Objects, FUQ, 032).

Interview analysis themes of 'Transition', 'Threshold', 'Epiphany' and 'Catalyst' all speak to the extent to which participants found the experience transformative: we can suggest that such a space is liminal for both the personal and the professional self. The data reveals that a wide range of emotions were experienced, including the uncomfortable: '*anxiety*', '*difficult*', '*raw*', '*fear*', '*disgust*'. Where disorientation and ambiguity are found, the ground can be fertile for the individual to transition through that liminal space towards the threshold of new understandings and concepts. If the latter are grasped then the discussion started in the workshops can act as a catalyst. It is clear from interviews that participants, stimulated by the case studies in the workshops, negotiated meaning and changes of meaning, some even experiencing personal epiphanies particularly around the paradoxes evident in life and death: 'the irony of spending life trying to escape from the inescapable fact of death'; and that 'the end is not the end'. One interviewee states '*The irony is it's like the inevitable, there is no escape from it yet it seems everyone, a lot of people seem to try to escape from talking about it*' (P6, p6, 473–475).

## Key finding 4: The workshops were a positive experience

The majority of participants (93%) said that they would recommend the workshop to others (Table 1G), indicating that it was a positive and valuable experience. The words selected to describe the workshop experience further support the value of the workshops (Table 2). The most frequently selected words by participants suggested that their workshop experience was cognitively stimulating (i.e. 'interesting' was selected 175 times, and 'thought-provoking' 174 times). Some selected words that described an emotional experience (i.e. 'moving' (n = 52) and 'enjoyable' (n = 102)). A high number indicated that the workshop was 'relevant' (n = 89) or 'worthwhile' (n = 126) (Fig 3). Few chose what might be seen as negative adjectives, with the words 'distressing', 'irrelevant' and 'irritating' selected just once each. 'Sad' was selected 11 times (perhaps surprisingly low given the topics under discussion).

The additional free text words that participants offered indicate the nuance and depth of their experiences: revealing; emotional; valuable; surprising; important; reflective; enlightening; instructive; deep; perspective shift; enabling; stimulating; poignant; challenging (x2);

**Table 2. Words selected to describe the workshop experience, Post-WQ.**

| Words selected to describe the workshop | Number of participants selecting each word. Participants could select multiple words |
|---|---|
| Interesting | 175 |
| Thought-provoking | 174 |
| Worthwhile | 126 |
| Enjoyable | 102 |
| Relevant | 89 |
| Moving | 52 |
| Sad | 11 |
| Distressing | 1 |
| Irrelevant | 1 |
| Irritating | 1 |
| Boring | 0 |
| Additional words added by participants: | challenging (n = 2); deep; emotional; enabling; insightful; important; instructive; perspective shift; poignant; reflective; revealing; surprising; too rushed (n = 2); valuable |

insightful. Again, these provide a mixture of intellectual and emotional responses to the workshops.

Few suggestions for improvement were given, with the exception of requests for longer sessions (also reflected in the free choice words 'too rushed', added by two participants). A balance was struck by the project team as part of our reflective approach: throughout the project, workshop materials were reduced to enable greater depth of engagement and reflection. It was felt that longer workshops would have had a negative impact on attendance/recruitment.

Some participants reflected that the workshops had helped them come to terms with their own bereavements. While the workshops were not intended as 'therapy', there were nonetheless therapeutic benefits for some individuals. One participant, for example, wrote that thinking about how past cultures dealt with death '*has helped me with my own recent losses and also helped me feel the importance of my work in a society where we hurry past such milestones*' (Age

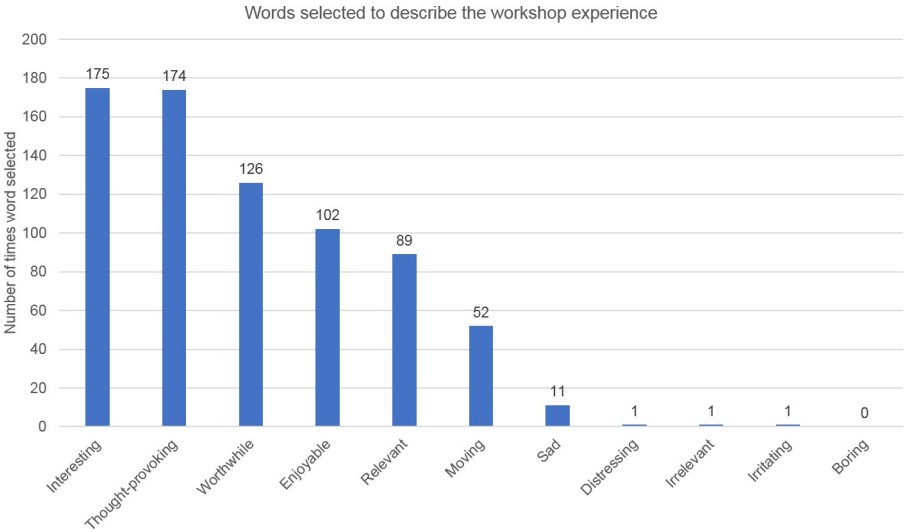

**Fig 3. Words selected by participants to describe the workshop (Post-WQ).**

and Circumstances of Death, FUQ, 19). Others reflected that '...*it helped in the long process of accepting death of myself and my loved ones*' (Age and Circumstances of Death, FUQ, 22); and that '*I think the workshop help[ed] me in continuing to be accepting of death*' (Images of the Dead, FUQ, 62); and finally, that '*People often do not want to talk about dying when it is going to happen to us all. Thinking about death and bereavement can be painful but I believe a lot of this pain could be lightened if more people attended these kind of workshops*' (Treatment of the Dead, FUQ, 108).

The emergent theme of 'Feelings' in the qualitative data is broad and much of it reflects positive experience of the workshops: 'stimulating'; 'enjoyable'; 'motivating'; 'comforting' and 'I am less scared'.

## Discussion

A high proportion of participants found that the workshops and content impacted upon their perceptions of death and bereavement, with 87% stating that the workshops had made them think differently about death and bereavement, and 93% stating they would recommend the workshop to colleagues. Whilst more than half of our participants had a lot of experience working with the dying or bereaved, they acknowledged that there were few opportunities to discuss their experiences and perspectives around death, even within palliative care professions and services. While a large proportion of individuals (84%) already felt confident discussing death and bereavement prior to the workshop, 47% anticipated an increase in confidence after participating, suggesting that even where confidence levels are high, there are significant benefits to confidence in discussing death and bereavement.

Most participants (92%) agreed that archaeology can be used as a tool for enabling discussion and challenging perceptions and biases around death, dying and bereavement. Although research considering the use of archaeology for modern day intervention and change is in its infancy, our findings are consistent with other literature which has effectively used the past to engage individuals in discussion about contemporary issues. This includes healthy eating [51] and psychological wellbeing [52] and calls for similar have been made in the field of climate change [53].

The results also demonstrate a self-reported impact on professional practice. Our findings demonstrate that for our participants, archaeology was an excellent way of starting conversations, both within the workshops themselves, and beyond, with a 'ripple effect' in personal and professional settings. The impact on student participants was very high: 24 students participated in workshops, with 3 attending more than one workshop. Of these, 79% (n = 19) thought that the workshop had made them think differently about death, dying and bereavement. Furthermore, whilst 79% of this group stated they were already confident in discussing death and bereavement, 42% agreed that their confidence increased as a result of workshop participation. These findings appear particularly pertinent considering the research described above regarding students' feeling ill-equipped to discuss death with dying patients [54].

Our results suggest that a number of factors were key in impacting the project participants, and Fig 4 presents a model of the mechanism for the successful delivery of this complex intervention.

Providing a safe environment was crucial for many participants. In addition to the detached or non-personal nature of the archaeological materials, a sense of safety came from coming together with the shared and explicit purpose of talking about death and dying. Meanwhile, shifts in attitudes towards death and bereavement were achieved through the creation of cognitive dissonance amongst participants. Cognitive dissonance [41]–a feeling of conflicting attitudes, beliefs and emotions–was created using archaeological case studies which challenged

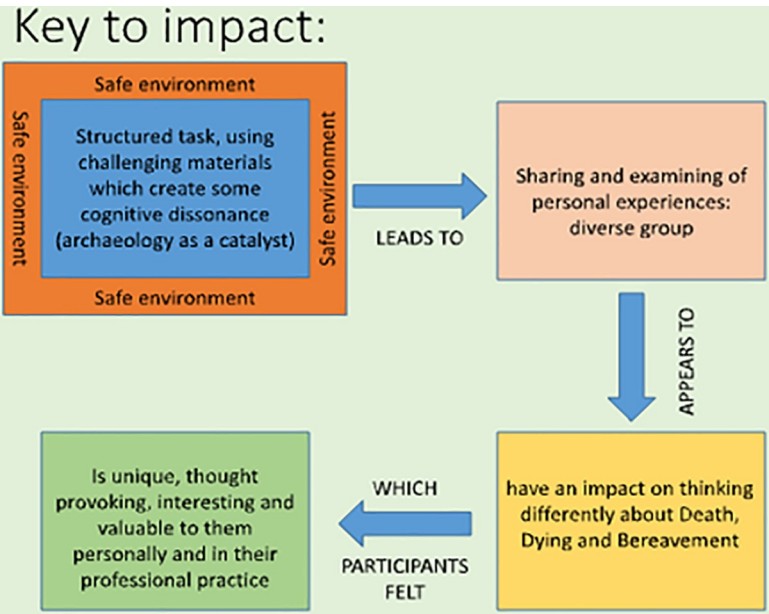

**Fig 4. Framework for the successful delivery of meaningful impact.**

perceptions of 'normal' behaviour. This dissonance led to the sharing and examination of personal experience, aided by group discussion by individuals from a range of backgrounds and with a diversity of personal experiences. This process helped participants to recognise and reflect upon their own subconscious cultural biases and move beyond individual idiosyncratic opinions, towards deeper understandings of the broader underlying themes behind many death practices. This is consistent with and supports our claim that for some participants, the workshops facilitated transition through a liminal space.

The use of case studies which participants linked to modern day practices helped to both challenge and normalise diverse funerary practices, enabling some participants to conceptualise more challenging practices through the lens of modern-day bereavement. This, in turn, encouraged reflection and empathy, and for some participants, suggested the humanity of grief spread across time and myriad cultures.

The analysis of free text comments suggested that personal changes in attitudes to death, dying and bereavement were necessary in creating impact in the professional lives of the participants, and that this shift in values, attitudes and beliefs will filter through to changes in practice over protracted periods of time. The dynamic and reciprocal nature of personal and professional identities is one that is recognised across disciplines such as nursing [55, 56] and medicine [57] and has given rise to the increasing emphasis on reflective practice [58] as a means of increasing awareness of how each impacts the other.

These findings highlight the value of interdisciplinary research [59]. Collaboration between archaeology, end of life care, nursing, and psychology is rare (this project is likely the first), but the approach has already inspired other projects [36, 60]. The results of the Continuing Bonds project add further weight to the argument that arts and humanities research can bring value to contemporary society '*by creating the conditions for change; . . . an openness, a space for experimentation and risk-taking at the personal, social and economic levels, [and] an ability to reflect in a safer and less direct way on personal, community and societal challenges*' [61, p153]. The Continuing Bonds project presents an example of how interdisciplinary and novel approaches can inspire change and create more resilient and compassionate societies.

## Limitations of the study

Despite actively advertising for participants of non-white British heritage, the demographic profile of our participants reflected broader trends in the non-medical health and social care workforce, in being predominantly white and female [62]. A greater diversity of participant backgrounds would be valuable in further testing the finding that cognitive dissonance, sharing personal experiences and norms, and constructive challenge is a valuable mediator of attitudinal change and development of practice. The interviews were conducted on a small sub-sample of participants who volunteered to be interviewed. The sample was therefore self-selecting, although did represent a number of sectors, including bereavement support, counsellors, nursing, end of life care professionals, and educators.

All but one participant attended the workshops voluntarily and were therefore prepared to engage in a new and innovative approach. Of the two participants who would not have recommended the workshop to colleagues, one had been required to attend by their line-manager, suggesting that voluntary participation may be important. Not only is voluntary participation a bedrock of ethical practice [63] research indicates that interventions geared towards change are more effective when they have been chosen by the individual themselves [64]. Further research would be needed to conclude whether the findings are transferable to others less interested or willing to participate in such workshops.

Few participants gave suggestions for improvement. The biggest issue was around the length of time individuals were given to engage with the archaeological case studies during the workshops, with some stating that more time would have been desirable. Though this would have been difficult to accommodate within the timeframe of the workshop, future studies could be of different lengths to sustain accessibility but allow some deeper exploration.

## Future work

The enhancement of knowledge and understanding of different practices, rituals and cultures was repeatedly acknowledged, leading to increased confidence and greater willingness to discuss death. This has clear potential as an important component in training, for both students and professionals.

### For students

Although our student sample was small, 63% (n = 15) agreed that the workshop would have an impact on how they approached death, dying and bereavement in their professional practice. Further research could seek to replicate our results in a larger student sample and focus on actual rather than anticipated change. Our results suggest that using materials from the archaeological record and from across the globe may have a particular value in nursing and medical training. If findings were to replicate our own, this would provide greater knowledge and a plurality of perspectives on death practices across time and space, as well as enabling deeper reflection, the challenging of biases and encouraging greater empathy around diverse practices.

### For professionals

While many health and social care professionals work with death on a daily basis, they receive little time to explore the topic in depth, including the historical background and multitude of practices of treating the dead. They also receive little time to explicitly reflect on their own experiences, values and beliefs in their professional environments. Taking time away from daily practices enabled our workshop participants to explore the topic in a safe space and

reflective atmosphere. The project findings are therefore valuable in shaping continuing professional development in this area, providing a dedicated space to reflect and share perspectives and experiences of caring for the dying and bereaved. One participant also recognised the relevance of the approach to psychotherapeutic work, which is another area for future research.

### Diversity training

The discussion of archaeological case studies, diverse practices and their relationship to culture may have transferability as a methodology to enhance the competence of staff in working with diversity and help to raise awareness of unconscious biases. As such, it would be valuable to explore the workshop methodology in equality and diversity training, which would also represent a new avenue for archaeological research and impact.

### Promoting and normalising discussion of death and bereavement

This study provides a novel approach to help normalise discussion around death and bereavement, as called for by the World Health Organisation, Living and Dying Well, and Dying Matters organisations [19–21]. Although the study targeted health and social care professionals and students, participants attended as individuals with their own personal experiences of bereavement. As has been demonstrated, the workshops transcended professional considerations, and often had personal impacts, with the materials prompting individuals to share stories about their loved ones and their own hopes, anxieties and expectations for themselves at end of life and in death. Our findings therefore align with other initiatives which seek to promote conversations around of death, dying and bereavement in the general population, such as the Dying Matters and Death Cafe movements [65, 66]. Since this study has shown that archaeology can successfully open up discussion around death, dying and bereavement, further research may fruitfully focus on the value of the same materials as catalysts for prompting discussion and reflection in different group settings, including for lay audiences. The follow-on project 'Continuing Bonds: Creative Dissemination' has explored the role of project findings for creative writing [67] and an additional pilot study recommends pursuing this approach for school groups to create compassionate communities for young people [68].

### Conclusions

The project has provided key insights into the value of archaeology for instigating conversations around death, dying and bereavement in health and social care professionals and students; in challenging biases and expectations; and serving as a catalyst for impact on professional practice. We believe that safe spaces, thought-provoking materials, and time for reflection were key to participant changes in values, attitudes and beliefs. While talking about death in the UK is not explicitly 'taboo' nor actively denied, the topic of death does remain, for many, troublesome and difficult, especially when death is imminent. The new approach explored here to enabling discussion and challenging biases is valuable, especially in an increasingly global and multi-cultural society.

Our findings demonstrate that for our participants, archaeology was an excellent way to start conversations, both within the workshops themselves, and beyond, as evidenced by the creation of a 'ripple effect' to colleagues, family and other social circles. The findings here suggest that archaeology can be a valuable tool in facilitating conversations and giving permission to discuss the difficult topics of death, dying and bereavement, and that these findings have value even amongst health and social care professionals, many of whom regularly encounter

death and dying in their professional lives and may be assumed to be more confident in discussing the topic.

Consideration of past death practices is also valuable in illuminating and challenging cultural biases and preconceived ideas about appropriate treatment of the dead and grieving practices, demonstrating the variety of responses and reactions to death and bereavement through time and space. This has important implications in the acceptance of diverse practices in our ever-increasing multi-cultural and multi-belief communities, and in fostering a culture of acceptance rather than judgement.

Through the 'deep time' perspective of archaeology, the workshops provided a shift in understandings of diverse death practices, with participants perceiving that this knowledge would go on to impact their professional practice. In particular, the participants emerged with a greater understanding of difference, were more accepting of practices and responses which diverge from those dictated by societal 'norms', and felt more comfortable and confident with the topic and their own perceptions of it. Fundamentally, the workshop led to personal reflections, bringing home the topics of death and dying, rather than the topics remaining distanced and abstract.

This interdisciplinary project has laid the foundation for informing nursing and health and social care education, training, and the wider profession, through normalising talk of the dead, and providing a new way forward to enhance communication, confidence, and understanding of this difficult but important topic.

## Supporting information

**S1 Table. Archaeological workshop case studies.**
(PDF)

**S1 File. Pre-Workshop Questionnaire (Pre-WQ).**
(PDF)

**S2 File. Post-Workshop Questionnaire (Post-WQ).**
(PDF)

**S3 File. Follow-Up Questionnaire (FUQ).**
(PDF)

## Acknowledgments

We are indebted to our workshop participants, as well as colleagues, professionals and contacts who aided in advertising and recruitment, including the Bradford District Care Trust, University Hospitals of Leicester NHS Trust, Leicestershire Partnership NHS Trust, and the Bradford Teaching Hospitals NHS Foundation Trust (particularly Jane Denison and Andrew Daley), and Marie Curie Hospice, Bradford. Thank you to Jane Booth, Eleanor Bryant, Mark Jones and Martin Truelove for their support in recruitment and promotion. Throughout the project, student volunteers have been invaluable to our work, and our thanks go to Haaroon Ahmed, Sakina Ali, Aiden Duday, Aoife Sutton and Stefan Yolov. Original project discussions were informed by Catherine Walshe, Neil Small, and Cary MacMahon. We thank our administrative and finance offices, ethics panels, and staff in research and knowledge transfer in LOROS and the University of Bradford, particularly Deborah Clarke, Richard Dunn, Omar Ali, Tamsin Holt, Martin Brinkworth, Sharon Mason, Idaliza Nukis and Emma Bowler. Many thanks to Christopher Gaffney as Head of Archaeological and Forensic Sciences at the University of

Bradford and for commenting on the paper, and to Zoe Edwards and Adrian Evans for also reading drafts of this paper; any oversights remain our own.

## Author Contributions

**Conceptualization:** Karina Croucher, Laura Green, Christina Faull.

**Data curation:** Jennifer Dayes.

**Formal analysis:** Jennifer Dayes, Justine Raynsford, Louise Comerford Boyes, Christina Faull.

**Funding acquisition:** Karina Croucher, Laura Green, Christina Faull.

**Investigation:** Karina Croucher, Lindsey Büster, Jennifer Dayes, Laura Green, Christina Faull.

**Methodology:** Karina Croucher, Lindsey Büster, Jennifer Dayes, Laura Green, Christina Faull.

**Project administration:** Karina Croucher.

**Resources:** Lindsey Büster.

**Supervision:** Karina Croucher.

**Validation:** Justine Raynsford, Louise Comerford Boyes, Christina Faull.

**Visualization:** Karina Croucher, Lindsey Büster, Jennifer Dayes.

**Writing – original draft:** Karina Croucher, Justine Raynsford.

**Writing – review & editing:** Karina Croucher, Lindsey Büster, Jennifer Dayes, Laura Green, Louise Comerford Boyes, Christina Faull.

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
