## [Decision Letter · Decision Letter 0]

13 May 2020

PONE-D-20-02889

Archaeology and contemporary death: Using the past to provoke, challenge and engage

PLOS ONE

Dear Dr Croucher,

Thank you for submitting your manuscript to PLOS ONE. After careful consideration, we feel that it has merit but does not fully meet PLOS ONE’s publication criteria as it currently stands. Therefore, we invite you to submit a revised version of the manuscript that addresses the points raised during the review process.

Please provide further evidence in your results to support the conclusions of your study.  As presented, descriptive statistics and a statement of growing recognition are not sufficient.

We would appreciate receiving your revised manuscript by 13 June 2020. To enhance the reproducibility of your results, we recommend that if applicable you deposit your laboratory protocols in protocols.io, where a protocol can be assigned its own identifier (DOI) such that it can be cited independently in the future. For instructions see: http://journals.plos.org/plosone/s/submission-guidelines#loc-laboratory-protocols

We look forward to receiving your revised manuscript.

Kind regards,

Rosemary Frey

Academic Editor

PLOS ONE

2. We noted in your submission details that a portion of your manuscript may have been presented or published elsewhere. ["Figures 2 and 3 were also used in Büster et al 2018 (preliminary results) - authors retain copywrite permission. "] Please clarify whether this [conference proceeding or publication] was peer-reviewed and formally published. If this work was previously peer-reviewed and published, in the cover letter please provide the reason that this work does not constitute dual publication and should be included in the current manuscript.

3. Please remove your figures from within your manuscript file, leaving only the individual TIFF/EPS image files, uploaded separately.  These will be automatically included in the reviewers’ PDF.

4. Please ensure that you refer to Figure 6 in your text as, if accepted, production will need this reference to link the reader to the figure.

Reviewers' comments:

Reviewer's Responses to Questions

**Comments to the Author**

1. Is the manuscript technically sound, and do the data support the conclusions?

Reviewer #1: Partly

Reviewer #2: Yes

2. Has the statistical analysis been performed appropriately and rigorously? 

Reviewer #1: N/A

Reviewer #2: I Don't Know

3. Have the authors made all data underlying the findings in their manuscript fully available?

Reviewer #1: Yes

Reviewer #2: No

4. Is the manuscript presented in an intelligible fashion and written in standard English?

Reviewer #1: No

Reviewer #2: Yes

5. Review Comments to the Author

Reviewer #1: General Comments

The inadequate referencing within the paper made it a challenge to stay focussed on your message, which is a shame because your topic is really very interesting and novel. It was confusing when you had some references in the reference list and others in the text. Also some claims you make need more than just one reference to support them, especially in the introduction where you are linking place of death with a lack of willingness to discuss.... E.g., This disconnect is in part due to a lack of willingness by public, health and social care professionals to focus on how the last phase of life should be discussed and negotiated, including considerations about key decisions regarding health care interventions such as admission to hospital and resuscitation. Also, the first reference in your list requires more details.

The paper requires a careful read to check whether statements in building your case for your research (in the background) are sufficiently supported, as well as in your conclusion (e.g. evidence for 'growing recognition'. The reference list needs a general tidy up.

Consider giving the bigger methods picture before honing in on how you chose your themes. It wasn't clear to me why I was suddenly reading about that.

What made the workshops successful? I don’t see this backgrounded in your introduction – it is starting to feel like a report. It is not clear to me where your results end and where your discussion begins. Also, this needs to relate more closely to your research question.

I suggest you divide up your results from your discussion so it is clear when you are making the transition and put in the separate headings. Also, add in recommendations section and limitations section.

Add more detail around the "Themes were coded at the semantic level, focusing on description rather than interpretation (see Braun and Clarke"

Abstract could be tighter. Watch repetition e.g. 'althoughs' in first sentence. What is BME in Figure 1?

Is 'white British' the standard descriptor for ethnicity in Britain?

Figure 2 & 4 could be presented using text. Pie chart isn't really needed for those data.

There are other mistakes in the script which require a careful edit. (e.g. word order error, run-on sentences, incorrect use of proper nouns.

Reviewer #2: This is an interesting project that uses archaeological case studies to provoke reflection among professionals engaged with the dying and dead in the UK in order to enable them to provide better support for more diverse communities in the UK. The premise is that death is believed to be a taboo subject in ‘modern western societies’, like the UK, and because of that belief, people are unprepared to adequately prepare for their own death and those of others. By familiarizing these professionals with other ways that people in the past treated their dead it is presumed that they would become more comfortable discussing death with their patients and clients (and be better able to engage with those clients and patients who have ‘different’ approaches to death). They found that workshops using archaeological materials - because of these materials’ ‘detached and safe’ qualities - provoked useful conversations and reflections, particularly among those professionals in their early career stage.

I cannot provide critical feedback on the analytical methods used in this paper, as I am unfamiliar with best practices in these kinds of social science studies. I do have a few thoughts on some of the underlying premises and framework of the paper.

1. This work follows in a long history of work by scholars, like Ernest Becker (cited by the authors) and Geoffrey Gorer (Gorer, G., 1955. The pornography of death. Encounter, 5(4), pp.49-52, not cited by the authors), who argue that westerners deny death (avoiding discussion/thinking about of it, etc.). Even though the authors recognize that these beliefs are contextual, they cite a work by Walter 1991, which is now nearly 30 years ago, I would be interested to know if, indeed, this notion holds true today in the UK, Europe, and North America, etc. Major shifts have occurred in the last 30 years, with the growing hospice movement, that makes me wonder if this taboo still exists. Are there any more recent studies of this attitude? I also wonder if this idea of ‘westerners treat death as a taboo/non-western peoples don’t have this idea’ is problematic in itself. The notion of western/non-western is problematic, and one could argue that such a dichotomy (and a focus on how ‘others do it’) perpetuates a kind of exoticization of unfamiliar peoples (unfamiliar to white British professionals, that is). See more on this here: Fabian, J., 1972. How others die—reflections on the anthropology of death. Social Research, pp.543-567.

2. Another aspect of this piece that needs to be addressed is the lack of adequate cultural contextualization/signposting. For example, in the abstract there is little specificity about where the Arts and Humanities Council is located, where the team of researchers are from (the UK), and where the workshop participants are from (UK). It is only on page four that we learn that: The team were based across universities and hospices in two cities in the North of England and the Midlands (but what characterizes this region/economically and culturally?).

3. The use of terms such as ‘modern, western society’ has been critiqued in recent years (see works on occidentalism and orientalism). Furthermore, it is problematic that the UK is framed as the proxy for contemporary/modern western society, but at the same time recognizes that it is a diverse place (with people engaged in diverse practices and holding different beliefs regarding death). If what the authors are signaling is that a ‘modern western society’ is culturally diverse, then that is what should be stated. I would suggest avoiding this term altogether (‘modern western society’).

4. Since archaeology plays such a central role in this study, it would be useful to know exactly what case studies were used. These should be listed in their entirety. Also how is archaeology being defined? It is confusing that (in one of the few examples mentioned in the text) Victorian death portraits were contrasted with family portraits taken with mummified body of living Sulawesi people (What is the ethnic/religious background of the people in this case study? Sulawesi is ethnically/religiously diverse). The ‘archaeological’ nature of this Sulawesi example is confusing, as the people being referred to are living.

6. PLOS authors have the option to publish the peer review history of their article (what does this mean?). If published, this will include your full peer review and any attached files.

Reviewer #1: No

Reviewer #2: No

---

## [Author Response · Author response to Decision Letter 0]

25 Sep 2020

Please note - comments below are also in the 'response to reviewers' file which has been uploaded. 

Dear Editor/reviewers

The Continuing Bonds team thank you for the thoughtful comments. We have taken these on board and believe this has resulted in a much more robust paper. In response to the review comments, we have substantially re-structured the paper. In addition to the reviewers’ comments we have added data to the results (such as the words used to describe the workshops), and made the findings clearer. We are incredibly grateful for the reviewers’ comments and hope you will find that the revised version meets the high standards for publication in Plos ONE. 

For clarity, the responses have been addressed in the table below: 

Comment: Response: Complete?

Please provide further evidence in your results to support the conclusions of your study. As presented, descriptive statistics and a statement of growing recognition are not sufficient. The structure has been changed to enable a clearer link between the study, results and conclusions, and a rewording of the conclusion to ensure we are drawing on the results. Thank you for pointing out this fundamental improvement. Y

General comments: 

 Manuscript and files have been checked and now conform with requirements. Y

2. We noted in your submission details that a portion of your manuscript may have been presented or published elsewhere. ["Figures 2 and 3 were also used in Büster et al 2018 (preliminary results) - authors retain copywrite permission. "] Please clarify whether this [conference proceeding or publication] was peer-reviewed and formally published. If this work was previously peer-reviewed and published, in the cover letter please provide the reason that this work does not constitute dual publication and should be included in the current manuscript. Thanks for noting this. The Büster et al 2018 paper was a snapshot of the project taken at the end of the first year, written for an archaeological audience. We confirm that the Plos ONE submission does not replicate this paper and has not been published elsewhere. The images that were reused in the 2018 publication have been removed and replaced by new, unused images. Y

 3. Please remove your figures from within your manuscript file, leaving only the individual TIFF/EPS image files, uploaded separately. These will be automatically included in the reviewers’ PDF. Images have been removed and uploaded separately. Y

4. Please ensure that you refer to Figure 6 in your text as, if accepted, production will need this reference to link the reader to the figure. Added (now Fig 4). Y

Reviewer #1 

Referencing: The inadequate referencing within the paper made it a challenge to stay focussed on your message, which is a shame because your topic is really very interesting and novel. It was confusing when you had some references in the reference list and others in the text. All references have been reviewed and updated. Y

Also some claims you make need more than just one reference to support them, especially in the introduction where you are linking place of death with a lack of willingness to discuss.... E.g., This disconnect is in part due to a lack of willingness by public, health and social care professionals to focus on how the last phase of life should be discussed and negotiated, including considerations about key decisions regarding health care interventions such as admission to hospital and resuscitation. Added in contemporary and more internationally relevant references. y

Also, the first reference in your list requires more details. Updated. Y

The reference list needs a general tidy up. Done Y

The paper requires a careful read to check whether statements in building your case for your research (in the background) are sufficiently supported, as well as in your conclusion (e.g. evidence for 'growing recognition'. This has been addressed – we structured the paper to make the findings, discussion and conclusion much clearer. Y

Consider giving the bigger methods picture before honing in on how you chose your themes. It wasn't clear to me why I was suddenly reading about that More thorough methods section has been added. Y

What made the workshops successful? I don’t see this backgrounded in your introduction – it is starting to feel like a report. This has been included more thoroughly. Y

 It is not clear to me where your results end and where your discussion begins. Also, this needs to relate more closely to your research question. I suggest you divide up your results from your discussion so it is clear when you are making the transition and put in the separate headings. Thank you for this suggestion – we have restructured, separating the findings and the discussion. 

 Y

Also, add in recommendations section and limitations section Limitations section added; Future work section added Y

Add more detail around the "Themes were coded at the semantic level, focusing on description rather than interpretation (see Braun and Clarke" This has been contextualised Y

Abstract could be tighter. Watch repetition e.g. 'althoughs' in first sentence. Abstract tightened (sorry, without tracked changes due to an error - we can attempt to get changes if needed). Y

What is BME in Figure 1? Black and Minority Ethnic - this figure has been removed Y

Is 'white British' the standard descriptor for ethnicity in Britain? Yes Y

Figure 2 & 4 could be presented using text. Pie chart isn't really needed for those data. These have been removed a table of data is presented instead (Table 1) Y

There are other mistakes in the script which require a careful edit. (e.g. word order error, run-on sentences, incorrect use of proper nouns. Complete Y

Reviewer #2 

This work follows in a long history of work by scholars, like Ernest Becker (cited by the authors) and Geoffrey Gorer (Gorer, G., 1955. The pornography of death. Encounter, 5(4), pp.49-52, not cited by the authors), who argue that westerners deny death (avoiding discussion/thinking about of it, etc.). Even though the authors recognize that these beliefs are contextual, they cite a work by Walter 1991, which is now nearly 30 years ago, I would be interested to know if, indeed, this notion holds true today in the UK, Europe, and North America, etc. Major shifts have occurred in the last 30 years, with the growing hospice movement, that makes me wonder if this taboo still exists. Are there any more recent studies of this attitude? I also wonder if this idea of ‘westerners treat death as a taboo/non-western peoples don’t have this idea’ is problematic in itself. The notion of western/non-western is problematic, and one could argue that such a dichotomy (and a focus on how ‘others do it’) perpetuates a kind of exoticization of unfamiliar peoples (unfamiliar to white British professionals, that is). See more on this here: Fabian, J., 1972. How others die—reflections on the anthropology of death. Social Research, pp.543-567. Thankyou for this really useful insight. We have undertaken further research and added in further supporting evidence for the difficulty around talking about death. 

We have also removed terms such as ‘western culture’ which we agree are problematic. Y

Another aspect of this piece that needs to be addressed is the lack of adequate cultural contextualization/signposting. For example, in the abstract there is little specificity about where the Arts and Humanities Council is located, where the team of researchers are from (the UK), and where the workshop participants are from (UK). It is only on page four that we learn that: The team were based across universities and hospices in two cities in the North of England and the Midlands (but what characterizes this region/economically and culturally?). We have made the abstract more concise, and included greater contextualisation in the introduction.

We did include the following information: ‘These two regions of the UK are characterised by relatively high rates of deprivation compared to the National average: Bradford ranks 11th in the UK; Leicester, previously ranked 23rd, has recently moved outside of the top 30.’ however, when we re-reviewed the text, this didn’t fit - this wasn’t the motivation behind selecting these study sites. 

 Y

The use of terms such as ‘modern, western society’ has been critiqued in recent years (see works on occidentalism and orientalism). Furthermore, it is problematic that the UK is framed as the proxy for contemporary/modern western society, but at the same time recognizes that it is a diverse place (with people engaged in diverse practices and holding different beliefs regarding death). If what the authors are signaling is that a ‘modern western society’ is culturally diverse, then that is what should be stated. I would suggest avoiding this term altogether (‘modern western society’).

 Agree - thank you for raising this, we have removed the term

 Y

Since archaeology plays such a central role in this study, it would be useful to know exactly what case studies were used. These should be listed in their entirety.

 Now included in additional files, S1 Table

 Y

Also how is archaeology being defined?

It is confusing that (in one of the few examples mentioned in the text) Victorian death portraits were contrasted with family portraits taken with mummified body of living Sulawesi people (What is the ethnic/religious background of the people in this case study? Sulawesi is ethnically/religiously diverse). The ‘archaeological’ nature of this Sulawesi example is confusing, as the people being referred to are living. Explanation added into section on archaeological materials Y

While revising your submission, please upload your figure files to the Preflight Analysis and Conversion Engine (PACE) digital diagnostic tool, https://pacev2.apexcovantage.com/. PACE helps ensure that figures meet PLOS requirements. To use PACE, you must first register as a user. Registration is free. Then, login and navigate to the UPLOAD tab, where you will find detailed instructions on how to use the tool. If you encounter any issues or have any questions when using PACE, please email us at figures@plos.org. Please note that Supporting Information files do not need this step. Complete Y

With thanks

Karina Croucher, Corresponding Author

---

## [Decision Letter · Decision Letter 1]

12 Nov 2020

PONE-D-20-02889R1

Archaeology and contemporary death: Using the past to provoke, challenge and engage

PLOS ONE

Dear Dr. Croucher,

Thank you for submitting your manuscript to PLOS ONE. After careful consideration, we feel that it has merit but does not fully meet PLOS ONE’s publication criteria as it currently stands. Therefore, we invite you to submit a revised version of the manuscript that addresses the points raised during the review process.

Please address Reviewer 2 comments concerning the inclusion of cultural context in the abstract.  Please also include further content from the workshops and address the discussion recommendations of Reviewer 3.

We look forward to receiving your revised manuscript.

Kind regards,

Rosemary Frey

Academic Editor

PLOS ONE

Reviewers' comments:

Reviewer's Responses to Questions

**Comments to the Author**

1. If the authors have adequately addressed your comments raised in a previous round of review and you feel that this manuscript is now acceptable for publication, you may indicate that here to bypass the “Comments to the Author” section, enter your conflict of interest statement in the “Confidential to Editor” section, and submit your "Accept" recommendation.

Reviewer #2: All comments have been addressed

Reviewer #3: (No Response)

2. Is the manuscript technically sound, and do the data support the conclusions?

Reviewer #2: Yes

Reviewer #3: Partly

3. Has the statistical analysis been performed appropriately and rigorously? 

Reviewer #2: I Don't Know

Reviewer #3: N/A

4. Have the authors made all data underlying the findings in their manuscript fully available?

Reviewer #2: Yes

Reviewer #3: Yes

5. Is the manuscript presented in an intelligible fashion and written in standard English?

Reviewer #2: Yes

Reviewer #3: Yes

6. Review Comments to the Author

Reviewer #2: The title and abstract still make no mention of the cultural/national context for this research. It is always important to situate and provide a geographic context for research on what are fundamentally cultural practices.

So, for example, in line 35: In this interdisciplinary pilot study, archaeological case studies were used in 31 structured workshops with 187 participants from health and social care backgrounds – it should be added – in the UK.

Reviewer #3: The new manuscripts submitted by the authors deals with most of the corrections presented by the peer reviewers, many of which deal with technical aspects of the work. However, the paper still presents a strong straw man argument where the front of the article argues that death is perceived as taboo and is not discussed particularly in a palliative care setting. And yet the subjects involved in these workshops indicated that they would have been happy discussing death with their family prior to the workshops. As a result, the article tends to show that the archaeology of death is interesting, rather than transforming perceptions around death experience. In part this can be addressed by the reframe implied by review 2s suggestions – who pointed about that Walter is an old source, and by exploring he context of this discussion a bit more – Becker’s fear of death hypothesis has been rather dramatically disproven, which suggests that we do not fully know the role of death in modern social context. Importantly it would be useful for this paper to present some of the contend of the workshops and explain how these created a discussion of mortality or dying specifically, as opposed to general conversations about commemoration or the apparent eccentricity of past peoples. An archaeology of dying is still not fully comprehensible so how did the cases facilitate that claim?

7. PLOS authors have the option to publish the peer review history of their article (what does this mean?). If published, this will include your full peer review and any attached files.

Reviewer #2: No

Reviewer #3: No

---

## [Author Response · Author response to Decision Letter 1]

30 Nov 2020

Dear Editor/reviewers

The authors thank you for the thoughtful comments. We have responded below to the additional comments raised by Reviewer 3 and hope the paper now meets the high standards for publication in Plos ONE. 

Reviewer's Responses to Questions

Comments to the Author

1. If the authors have adequately addressed your comments raised in a previous round of review and you feel that this manuscript is now acceptable for publication, you may indicate that here to bypass the “Comments to the Author” section, enter your conflict of interest statement in the “Confidential to Editor” section, and submit your "Accept" recommendation.

Reviewer #2: All comments have been addressed

Reviewer #3: (No Response)

>> Thank you, we are pleased that reviewer 2 recognises that all comments have been addressed

2. Is the manuscript technically sound, and do the data support the conclusions?

Reviewer #2: Yes

Reviewer #3: Partly

>> We are delighted that Reviewer 2 is happy the data support the conclusions, however, to further address the concerns of reviewer 3, we have added in additional data which further supports our claims (see detailed response below for line numbers)

3. Has the statistical analysis been performed appropriately and rigorously?

Reviewer #2: I Don't Know

Reviewer #3: N/A

>> No response required

4. Have the authors made all data underlying the findings in their manuscript fully available?

Reviewer #2: Yes

Reviewer #3: Yes

>> No response required

5. Is the manuscript presented in an intelligible fashion and written in standard English?

Reviewer #2: Yes

Reviewer #3: Yes

>> No response required

6. Review Comments to the Author

Reviewer #2: The title and abstract still make no mention of the cultural/national context for this research. It is always important to situate and provide a geographic context for research on what are fundamentally cultural practices.

So, for example, in line 35: In this interdisciplinary pilot study, archaeological case studies were used in 31 structured workshops with 187 participants from health and social care backgrounds – it should be added – in the UK.

>> Addressed, added to line 38 in the abstract, and line 147 in the text. 

Reviewer #3: The new manuscripts submitted by the authors deals with most of the corrections presented by the peer reviewers, many of which deal with technical aspects of the work. However, the paper still presents a strong straw man argument where the front of the article argues that death is perceived as taboo and is not discussed particularly in a palliative care setting. And yet the subjects involved in these workshops indicated that they would have been happy discussing death with their family prior to the workshops. As a result, the article tends to show that the archaeology of death is interesting, rather than transforming perceptions around death experience. In part this can be addressed by the reframe implied by review 2s suggestions – who pointed about that Walter is an old source, and by exploring he context of this discussion a bit more – Becker’s fear of death hypothesis has been rather dramatically disproven, which suggests that we do not fully know the role of death in modern social context. Importantly it would be useful for this paper to present some of the contend of the workshops and explain how these created a discussion of mortality or dying specifically, as opposed to general conversations about commemoration or the apparent eccentricity of past peoples. An archaeology of dying is still not fully comprehensible so how did the cases facilitate that claim?

 >> Thank you for your observations here. We have worked extensively to address the issues raised by reviewer 3 in the following ways: 

1) We have revised (toned down) the language around the concept of ‘taboo’ and rephrased as ‘troublesome’ and/or ‘problematic’. E.g. in lines: lines 80-84; 831-833; 839-842

2) Additional literature has been added around the modern social context of death: e.g. lines 71-75; 91-102; 111-113. These additional sources also add nuance around the idea that there are differences in attitudes towards death in general/abstracted, and personal death/planning for death

3) To further support our argument that the workshops have had significant impacts on the participants, we have added additional interview data which highlights participant responses to the material, including their use of the word ‘taboo’: Lines 400-403; 486-513; 551-560; 604-617; 659-661; 867-69

Relevant information about the insights from interviews have been added throughout, i.e. 297-98; 335-342; 348-350; 358-9; 410-11; 746-49

An additional author has been added, who has analysed the interview data (and had already commented on earlier drafts) cementing her role as a co-author. 

4) Although many of the workshop participants were already comfortable talking about death, the section starting on line 447 talks about participant comfort around death, and further figures have been added to make it clearer that the 58% saying they were already comfortable talking about death, was 58% of those who ‘disagreed’ rather than 58% of the whole (12% of respondents) – lines 443-444 – hopefully this makes it clearer and is less likely to misrepresent, and makes clear that our argument is not a ‘straw man’, but based on our data, from b1oth participants, and in the literature discussed above. We also draw attention that while participants still have confidence around death, they still reported a positive change from participating in the workshops. We believe the data and added nuance address the concerns raised by Reviewer 3.

5) Additional archaeological examples have been added in: i.e. lines 225-247; 268-270. These have been kept fairly brief, as adding further archaeological material would move the paper in a different direction, and is the subject of an earlier paper (Büster et al 2017, see line 220-21), and of forthcoming work. To pinpoint the exact case studies and how they inspired a reaction would require detailed analyses of conversations within the workshops, which is outside of the scope of this paper, and while interesting, our observations from being in the workshops suggest that different individuals responded differently to different case studies. We intentionally have not talked about ‘an archaeology of dying’ in this paper – the case studies present different reactions to death through time and place, rather than explicitly discussing ‘dying’ through time. To further explore the individual archaeological case studies and individual workshop reactions is beyond the scope of this paper. This paper presents a wider perspective on the impacts of the workshop and its material.

Thank you for your time in reviewing the paper. This paper has been delayed heavily due to the pandemic and we would be grateful for a timely response if at all possible (although we realise how busy you will be). 

With thanks

Karina Croucher, Corresponding Author

---

## [Decision Letter · Decision Letter 2]

3 Dec 2020

Archaeology and contemporary death: Using the past to provoke, challenge and engage

PONE-D-20-02889R2

Dear Dr.Croucher,

We’re pleased to inform you that your manuscript has been judged scientifically suitable for publication and will be formally accepted for publication once it meets all outstanding technical requirements.

Kind regards,

Rosemary Frey

Academic Editor

PLOS ONE

Additional Editor Comments (optional):

Reviewers' comments:

Reviewer's Responses to Questions

**Comments to the Author**

1. If the authors have adequately addressed your comments raised in a previous round of review and you feel that this manuscript is now acceptable for publication, you may indicate that here to bypass the “Comments to the Author” section, enter your conflict of interest statement in the “Confidential to Editor” section, and submit your "Accept" recommendation.

Reviewer #2: All comments have been addressed

Reviewer #3: All comments have been addressed

2. Is the manuscript technically sound, and do the data support the conclusions?

Reviewer #2: Yes

Reviewer #3: Yes

3. Has the statistical analysis been performed appropriately and rigorously? 

Reviewer #2: N/A

Reviewer #3: N/A

4. Have the authors made all data underlying the findings in their manuscript fully available?

Reviewer #2: Yes

Reviewer #3: Yes

5. Is the manuscript presented in an intelligible fashion and written in standard English?

Reviewer #2: Yes

Reviewer #3: Yes

6. Review Comments to the Author

Reviewer #2: The authors have addressed my concerns (and those of the other reviewer). I agree with the other reviewer that while the participants may not have seen death as a taboo prior to the study, they appear to have benefited from the knowledge of diverse mortuary practices.

Reviewer #3: Much improved, the article now address the points well and frames the taboo topic in light of the project parameters, and it is good to see the case studies explained in detail.

7. PLOS authors have the option to publish the peer review history of their article (what does this mean?). If published, this will include your full peer review and any attached files.

Reviewer #2: No

Reviewer #3: No

---

## [Editor Report · Acceptance letter]

16 Dec 2020

PONE-D-20-02889R2 

Archaeology and contemporary death: Using the past to provoke, challenge and engage 

Dear Dr. Croucher:

I'm pleased to inform you that your manuscript has been deemed suitable for publication in PLOS ONE. Congratulations! Your manuscript is now with our production department. 

Kind regards, 

on behalf of

Dr. Rosemary Frey 

Academic Editor

PLOS ONE